

# Analysis of the performance of a ship-borne scanning wind lidar in the Arctic and Antarctic

Rolf Zentek, Svenja H. E. Kohnemann, and Günther Heinemann

Department of Environmental Meteorology, University of Trier, Germany

**Correspondence:** Rolf Zentek (zentek@uni-trier.de)

**Abstract.** Profiles of wind speed and direction at high spatial and temporal resolution are fundamental meteorological quantities for studies of the atmospheric boundary layer. Ship-based Doppler lidar measurements can contribute to fill the data gap over oceans particularly in polar regions. In the present study a non-motion stabilized scanning Doppler lidar was operated on board of RV *Polarstern* in the Arctic (June 2014) and Antarctic (December–January 2015/2016). This is the first time that

such a system measured on an icebreaker in the Antarctic. A method for a motion correction of the data in the post-processing is presented. The wind calculation is based on vertical azimuth display (VAD) scans with eight directions that pass a quality control. Additionally a method for an empirical signal-to-noise ratio (SNR) threshold is presented, which can be calculated for individual measurement setups. Lidar wind profiles are compared to total of about 120 radiosonde profiles and also to wind measurements of the ship.

The performance of the lidar measurements in comparison with radio soundings shows generally small RMSD (bias) for wind speed of around 1 m s$^{-1}$ (0.1 m s$^{-1}$) and for wind direction of around 12° (6°). The postprocessing of the non-motion stabilized data shows a comparable good quality as studies with motion stabilized systems.

Two case studies show that a flexible change of SNR can be beneficial for special situations. Further the studies reveal that short-lived Low-Level Jets in the atmospheric boundary layer can be captured by lidar measurements with a high temporal

resolution in contrast to routine radio soundings. The present study shows that a non-motion stabilized Doppler lidar can be operated successfully on an icebreaker. It presents a processing chain including quality control tests and error quantification, which is useful for further measurement campaigns.

## 1   Introduction

Changes in the Arctic and Antarctic climate system are strongly related to atmosphere-ocean-ice (AOI) interactions and feed-

backs between the atmospheric boundary layer and the free atmosphere. Hence, the knowledge about the state of the atmospheric boundary layer (ABL) is crucial for the understanding of AOI processes, atmospheric transports, air pollution processes and the verification and improvement of numerical weather forecast and climate models for polar regions. Profiles of wind speed and direction at high spatial and temporal resolution are fundamental meteorological quantities for ABL studies. While at mid-latitudes the ABL is studied using tall towers and ground-based remote sensing instruments such as lidar, radar or sodar

at several observatories, these measurements are rare or absent in the Arctic and Antarctic. Thus radiosondes are generally the





main source to measure quantities of the ABL in the polar regions. Since the radiosonde stations are primarily located over land, there are huge data gaps over the ocean. Furthermore, the temporal resolution of radio soundings is generally of the order of a couple of hours. Over the polar oceans, only few research vessels provide radio soundings, which are very valuable to improve the initial conditions for numerical weather forecasts and for reanalyses (e.g., Dee et al., 2011), but are insufficient for

detailed studies of boundary layer processes.

Ship-based Doppler lidar measurements are a possibility to fill the gap of radio soundings over oceans, since they provide wind profiles with a high spatial and temporal resolution (Tucker et al., 2009; Achtert et al., 2015). In addition, Doppler wind lidar measurements allow for the determination of the turbulence structure of the ABL (Banta et al., 2006; Pichugina et al., 2012; Kumer et al., 2016). If two Doppler lidars are available, techniques like the 'virtual tower' can be applied (Calhoun

et al., 2006; Damian et al., 2014). In synergy with additional remote sensing instruments measuring the temperature profile, the turbulent mixing conditions in the ABL can be described at high temporal and vertical resolution of 10 min and 10 m, respectively (Brooks et al., 2017).

In this study we analyze data from a scanning Doppler lidar on board of RV *Polarstern* in the Arctic (June 2014) and Antarctic (December–January 2015/2016). There are two important aspects measuring with a Doppler lidar on board of a moving ship

in polar regions: a) the ship's movement requires data corrections regarding its orientation and b) the adaptation of lidar measurement settings and analysis configuration for conditions with low backscatter due to the low aerosol concentration. Some studies present measurement campaigns dealing with challenge a) (e.g., Pichugina et al., 2012; Tucker et al., 2009; Achtert et al., 2015). All of them use a motion-stabilization platform to remove the effects of ship motion. We present a different option to deal with the varying orientation of the ship. The adaptation of measurement settings for the polar environment (challenge

b)) is less documented. The goal of these adaptions is the improvement of the signal-to-noise ratio (SNR). Hirsikko et al. (2014) recommend the use of an optimized telescope focal length of the lidar and the increase of the integration time for measurements in Finland. The main goal of the present paper is the assessment of the wind lidar performance in comparison with radiosondes on the German icebreaker *Polarstern*. A similar study was made by Achtert et al. (2015), who used a motion-stabilized scanning wind lidar during a cruise of the Swedish icebreaker ODEN in the Arctic in 2014 (Tjernström et al., 2014). Their three-month

campaign started immediately after our Arctic campaign in 2014. No ship-based measurement campaign of a Doppler wind lidar is known for the Antarctic. The combination of the measurement framework and the presented comprehensive analysis of the settings serve as basis for improvements in further data collections. The outline of the paper is as follows: In Section 2 an overview of the measurement campaigns and the data processing is given. Section 3 presents the results for intercomparisons of lidar data with radiosondes and ship wind measurements. Two case studies are shown in Section 4. A summary and conclusions

are given in Section 5.

## 2   Measurements and data processing

The measurements were performed during the two *Polarstern* cruises PS85 and PS96 of the Alfred-Wegener Institute Bremerhaven (Germany). The cruises with approximate sea ice conditions during the measuring periods are shown in Figure 1. PS85





took place in the Arctic from the 06th June till 03rd July 2014 and PS96 in the Antarctic from the 06th December 2015 till 14th February 2016. Lidar measurements were taken for a period of 18 days (12th till 29th July) during PS85 and for 38 days (24th December till 30th January) during PS96. *Polarstern* is the German research icebreaker with a length of 118 m and a weight of 17300 tons (Fig. 2). The typical cruise speed is 12 knots.

## 2.1 Doppler wind lidar

The instrument is a "Halo-Photonics Streamline" wind lidar, which is a scanner and can operate with a maximum range of 10 km, but was used only for a range up to 3600 m due to the low aerosol concentration (Table 1). The lidar was installed on the port (starboard) side of the ship during PS85 (PS96) approximately 20 m above the waterline (see Figure 2). Besides the lidar, an external Altitude Heading Reference System (AHRS; XSENS MTi-G-700-GPS/INS) was installed for higher frequency (sampled with up to 400 Hz) recordings of the ship's pitch and roll, in addition to lower frequency (1 Hz) navigation data from the ship's internal systems.

A variety of different scanning programs were used: vertical azimuth display (VAD), horizontal stare in two or three directions, range-height indicator (RHI) and vertical stare. In the present paper we will focus on the VAD measurements that allow the computation of vertical profiles of horizontal wind speed. One VAD scan is composed of eight rays with fixed elevation and different azimuth (0°, 45°, 90°, 135°, 180°, 225°, 270°, 315°). During PS96 we changed the elevation from 85° to 75° after three days. The averaging time for each ray was usually 12–15 seconds. During PS85 the averaging time for each ray was only 1.5 seconds but azimuth-circles were done at 25°, 50° and 75° elevation. For the analysis we will either use only the 75° or all 25°, 50° and 75° elevations. To make them comparable in case of using all three elevations, we will count the $3 \times 8 = 24$ rays as one VAD. One ray is divided into gates of 3 m length and the measured Doppler velocity is representative for six gates (18 m). During PS85 those six gates were non-overlapping, thus measurements were available every 18 m. During PS96 the six gates were overlapping, thus measurements were available every 3 m. VAD wind profiles are typically available every 15 minutes and a whole VAD scan required about 2 minutes for PS96. Photos of the weather condition were taken manually for special situations during PS85 and automatically with a GoPro (with constant power connection) every minute during PS96.

## 2.2 Radiosondes

Radiosondes at *Polarstern* (König-Langlo, 2014a, 2016a) were usually launched twice a day at 05 and 11 UTC during PS85 (39 radiosondes over the 18 days) and 07 and 11 UTC during PS96 (70 radiosondes over the 38 days). Radiosondes of the type Vaisala RS92 (Vaisala, 2013) were used. The measurement uncertainty for wind is specified as 0.15 m s$^{-1}$ for speed and 2° for direction. For the intercomparison of lidar wind profiles with the radiosonde profiles additional aspects apart from instrumental errors have to be considered. As shown below, the vertical range of the lidar is generally limited to the height of the ABL of a few hundred meters. When the ship is cruising, the radiosondes are launched close to the ship's superstructure and are affected by the turbulent wake of the ship. The radiosonde also needs time to accelerate to the ambient wind speed after launch, and exhibits strong pendulum motions during this phase. This results in a strong noise in the raw wind data, and a low-pass filter is applied, resulting in a reduced vertical resolution (estimated as about 200 m by Päschke et al. (2015)). As documented





by Achert et al. (2015) for the RV Oden, the ship superstructure modifies the mean flow depending on flow direction. The largest effect occurs for a relative wind along the ship's axis. For these conditions, the disturbance decreases with height and is estimated as smaller than 2% for horizontal wind speeds at altitudes above 75 m. For a flow being perpendicular to the ship this effect reduces to 2% also below 75 m. A study of Berry et al. (2001) for RV *Polarstern* shows that the largest flow distortion

for the ship orientated into the wind occurs as a wind decrease up to 30% in the lee of the main superstructure in the lowest 50 m (where the radiosonde is launched).

## 2.3 Analysis of the lidar data

The wind analysis consists of different steps. First we look at the influence and correction of the ship's motions. In the second part we describe our data processing method and computation of horizontal winds. In the third part we discuss our choice of

the signal to noise ratio threshold.

### 2.3.1 Ship motion correction

The main difficulty in receiving reliable wind data results from the movements of the ship. The ship's velocity and orientation and their changes influence the directions of the lidar's outgoing and incoming rays. Therefore the ship's velocity and orientation angles are the two main factors for the correction of the measured data. During both cruises PS86 and PS96, the ship

was moving with more than 1 m s$^{-1}$ about 50% of the time. The lidar was aligned with the ship by eye as best as possible (deviations of the yaw angle between lidar and the ship are discussed later in the results section). Measured ship data from the scientific navigational platform are taken to correct each single lidar measurement by the ship's speed and roll-pitch-yaw angles. The resolution of these data is 1 Hz. The correction for the ship's roll and pitch movements can be avoided by using a motion-stabilizing platform (Achert et al., 2015). We had no such platform, but additionally to the ship's 1 Hz navigation

data, we recorded roll and pitch movements also at high-frequency (up to 400 Hz) by the AHRS that was attached to the lidar. The AHRS data were used to determine the roll and pitch offset between the AHRS (resp. lidar) reference system and the ships reference system. During PS96 the averaging time of a single ray was typically 12–15 seconds, so that we corrected each single measurement with the mean value over the averaging time. All measurements that have a standard deviation of roll or pitch angle larger than 0.5° or yaw angle larger than 2° over this averaging time were excluded from the analysis in order to reduce

the error. It should be noted that the correction and filtering process causes almost no loss of data, as the ship's movement even during ice breaking conditions generally does not result in high-frequency changes of roll and pitch (except some cases of ramming). The important part is in fact the low-frequency change in roll and pitch (e.g. pumping water from one tank to another, changing cargo) that gets corrected. This can be seen by subtracting a 2-minute running median from the roll and pitch data (Fig. 3). The remaining angles are within -0.1° and 0.1° in 60–70% of the time. For a data point in 1 km distance from the

lidar a change of elevation from 75° to 75.5° (25° to 25.5°) causes a difference in height of 2 m (8 m) and a horizontal wind speed error of less than 3.3% (0.4%). This is acceptable as we will later interpolate over height intervals of 50 m. Without roll and pitch correction, values amount to -2° to 2° for roll (for 95% of the cases) and 0° to 1.5° for pitch. Therefore a setup without any roll or pitch correction at all would still provide usable data, if a high data quality is not of importance. For example, for a





data point in 1 km distance from the lidar a change of elevation from 75° to 77° (25° to 27°) causes a difference in height of 8 m (31 m) and horizontal wind speed error of less than 13% (17%). We also corrected for the influence of the angular velocity of roll pitch and yaw, but it was found to be negligible. For PS96 (PS85) the correction due to angular velocity was less than 0.2 m s$^{-1}$ for 99.7% (99.9%) of the time and never greater than 0.5 m s$^{-1}$.

### 2.3.2  Data processing

First a signal-to-noise ratio (SNR) threshold was chosen and all data points within one ray with a worse SNR were removed. Furthermore the first data points near the lidar were removed (approx. the first 30 m) as these measurements are often affected by the outgoing pulse. Then each single ray was segmented into bins of 100 m. For each bin, outliers (radial velocity > 3 × standard deviation) were removed. If less than 50% of the data remained or if the standard deviation of the radial velocity of
remaining data in the bin was greater than 3 m s$^{-1}$, the whole bin was removed. To compute a vertical profile of horizontal wind speed from a complete VAD we first divided all data points into layers of different heights. A thickness of 50 m was chosen for each layer for the radiosonde comparison, but thicknesses down to 10 m were tested as well. We used the standard assumption for VAD processing that the wind field is horizontally homogenous in each layer. The general approach for the processing of VAD scans is the calculation of the 3D wind by finding the solution to a system of equations. There are two
common perspectives on their definition. The first perspective operates in the (local) Cartesian coordinate system "East, North, Up", where wind is described by the components $(u, v, w)$ and the direction of the lidar beam (normalized radius vector $(x_L, y_L, z_L)$). Each measured Doppler velocity $d$ (negative, if wind is blowing towards the lidar) satisfies the following linear equation

$$d = x_L \cdot u + y_L \cdot v + z_L \cdot w \tag{1}$$

The second perspective describes wind with as horizontal wind speed and direction and the vertical component (ff = wind speed, dd = wind direction, $w$). The Doppler velocity is then a function of the scanning directions in polar coordinates ($\phi$ = azimuth, $\theta$ = elevation).

$$d = \cos(\phi - dd - \pi) \cdot ff \cdot \cos(\theta) + \sin(\theta) \cdot w \tag{2}$$

As equation 1 can be transformed into equation 2, they are equivalent (see appendix). Assuming that the lidar remains stationary
and has a fixed elevation $\theta$, the equation further simplifies to

$$\frac{d}{c_1} = \cos(\phi - dd - \pi) \cdot ff + w \cdot c_2 \tag{3}$$

with the constants $c_1 = \cos(\theta)$ and $c_2 = \tan(\theta)$. Wind speed and direction can then be determined by a cosine fit for all avalaible scan directions. Although the equations 2 and 3 are more intuitive, and our lidar software already uses the parameters elevation and azimuth, we found it is easier to work in a Cartesian coordinate system to apply corrections and thus choose
equation 1. Since we have eight rays per VAD (and more than one measurement per ray in each layer) we get a system of linear





equations. Given a measured set of Doppler velocities $d_i$ ($i = 1, ..., n$) in directions $(x_i, y_i, z_i)$ (east, north, up) the wind speed $(u, v, w)$ can be calculated by solving the overdetermined system

$$
\begin{pmatrix}
x_1 & y_1 & z_1 \\
x_2 & y_2 & z_2 \\
... & ... & ... \\
x_n & y_n & z_n
\end{pmatrix}
\times
\begin{pmatrix}
u \\
v \\
w
\end{pmatrix}
=
\begin{pmatrix}
d_1 \\
d_2 \\
... \\
d_n
\end{pmatrix}
\tag{4}
$$

using the least squares method. To ensure the quality of the data we added the condition that at least six out of eight azimuth
angles had data (that was not removed), thus at least measurements in a sector of 270° were available.

In Figure 4 we show the amount of computed wind speed / direction data from VAD scans for different SNR thresholds. The increase of computed data stagnates around -30 dB. A further decrease of the SNR threshold adds only data that is thrown out again by the "100m bin method" or for other reasons. One can also see the zig-zag artefact that is produced by this 100m-bin combined with computing winds every 50 m. It its more dominant in case of PS96 as the measurement was taken every 3 m
while the measurements for PS85 were taken every 18 m. The benefit of using additional scans with 25° and 50° elevation for PS85 can be seen for the lowest 750 m, if a higher SNR threshold is chosen. The choice of SNR threshold for this paper is explained in the next section.

### 2.3.3   Choice of signal-to-noise ratio thresholds

SNR-based thresholds for the separation between reliable and unreliable data points are a common technique for lidar data
processing (Päschke et al., 2015; Pearson et al., 2009; Frehlich and Yadlowsky, 1994; Barlow et al., 2011). This value can vary depending on the instrument specific performance (detector noise) and the variability of atmospheric conditions within the measured volume. The recommendation of the manufacturer for the lidar is -18.2 dB. However, Päschke et al. (2015) showed that this value is rather conservative and reduces the amount of data by up to 40% (between -20 dB and -18.2 dB). Hirsikko et al. (2014) use a threshold of -21 dB and state that -25 dB could still suitable for horizontal wind measurements.
Pearson et al. (2009) find experimentally a threshold SNR for reliable data of 23 dB. The potential SNR threshold was already considered during our measurements by adjusting telescope focal length of the lidar and the integration time (following the recommendations Hirsikko et al. (2014)). This is necessary during the measurements, since raw data of single pulses are not stored and no postprocessing is possible. Figure 4 shows the sensitivity of available data for PS85 and PS96 on the SNR threshold. We find a similar reduction as Päschke et al. (2015). A rule of thumb for our measurements seems to be increasing
the SNR threshold by 1 dB results in a (relative) loss of 5–10% of the data. Additionally due to the different averaging time for each ray during PS85 and PS96 (1.5 vs 12–15 sec), the PS96 data contains less noise and thus it makes sense to choose a different SNR threshold for each data set. Päschke et al. (2015) checked the measured wind speed of vertical stares. Knowing that these had to be around 0 m s$^{-1}$, the influence of noise could be evaluated. We follow a similar approach and evaluated the Doppler velocity from all individual rays for VAD scans with an elevation of 75° (only the first data points near the lidar
were removed; see subsection data processing). Since the Doppler velocity due to horizontal wind speed is less than 26% at this



elevation, the range of realistic Doppler velocities should be $\pm 10 \, \mathrm{m \, s^{-1}}$. Data points outside this range can be regarded as noise. This condition is used to find a SNR threshold in a three-step procedure. First, we look at the overall frequency distribution of measured Doppler velocities (Fig. 5, top). We assume that the data mainly consists of two parts: the noise (homogenous along all wind speeds; top to bottom) and the wind signal (relatively homogenous along the signal intensity or SNR; left to

right). Signal intensity is defined as SNR+1. All points above $10 \, \mathrm{m \, s^{-1}}$ or below -10 $\mathrm{m \, s^{-1}}$ are taken to construct a noise distribution as a function of intensity using the mean value (Fig. 5, bottom). We call this the empirical noise. In the second step, we take the ratio of the empirical noise and the mean of the measured Doppler velocities for each intensity, which results in an empirical noise fraction (plotted as solid line in Fig. 5, bottom). The noise fraction is close to zero for high intensities and starts to increase rapidly at different SNR values for both data sets. We choose a SNR threshold (step three) of -17 dB for PS85

and -20 dB for PS96. This empirical SNR threshold results in about 14%/26% of usable raw data for PS85/96. Comparing this to the resulting VAD percentages 14%/21% (Fig. 4) it should be noted that the decrease for PS96 comes mostly from the restriction sd(yaw) <2° and sd(roll/pitch)<0.5°. Without this condition, the computed VAD percentage is 25%.

## 3   Results

A verification of the lidar wind data is presented in the following by comparisons with radiosondes and ship measurements.

For the statistics of wind direction the absolute values of the differences are adjusted to be smaller than 180° to avoid the discontinuity at northerly directions (e.g. a difference of 270° becomes -90°). For the correlation of wind direction we used the correlation coefficient for angular variables (Jammalamadaka and Sarma, 1988). Radiosonde data was interpolated linearly with height to match the lidar data. Lidar wind speed and direction was first computed for every VAD and then averaged over a 20 min interval centered around the launch time (plus 100 sec) of the radiosonde (100 sec after the start the radiosonde is at a

height of around 500 m). We excluded all data points with wind speed < 0.5 $\mathrm{m \, s^{-1}}$ for the statistics of wind direction, but this condition was only met during PS96 and only for up to six data points at different heights/times. Figure 6 shows the calculated RMSD and bias by height for different SNR thresholds. While 23 dB leads to some larger differences particularly for PS85, our empirical thresholds of 20 dB and 17 dB are found to be reasonable. Furthermore, a systematic dependence on height is not present. At heights above 1000 m the sample size is relatively small and differences between different SNR thresholds are

not robust.

The overall statistics of the radiosonde comparisons is shown in Table 2. Although our data set is smaller than that of Achtert et al. (2015) we find similar results (RMSD for wind speed around 1 $\mathrm{m \, s^{-1}}$ and wind direction around 10°) except for our larger bias in wind direction. This bias persists even when applying a stricter condition for the allowed standard deviation of yaw angle during the measuring/averaging time (last row in Table 2). The bias for the wind speed is very small.

In order to quantify the impact of changes in our standard data processing, the effects of changing the layer thickness and changing the averaging time around the radiosonde launch were investigated. Table 3 summarizes the ranges of these effects RMSD, bias and $R^2$. None of these changes had any relevant influence. We also computed the 95% confidence interval bounds for the bias for wind direction which was found to be only up to 1° higher/lower than the biases given in Tables 2 and 3.





As mentioned above, our results are similar to Achert et al. (2015), who used a motion stabilized platform and found mean bias for wind of 0.3 m s$^{-1}$, and a mean standard deviation of 1.1 m s$^{-1}$ and 12° for wind speed and direction, respectively. However, our bias for wind direction is larger than the value of 2° found by Achert et al. (2015). As described in the section 2, the lidar was aligned with the ship's axis only by eye. We tried to estimate the yaw offset by checking the correlation of the

roll and pitch 1 Hz data from the AHRS (resp. lidar) and the ship navigation system. By assuming a yaw offset and correcting the roll and pitch angles, we determined the peak of the correlation. The results depend largely on the chosen time window and scattered between -5° to 5°. Overall, this can explain a lidar yaw offset of around -0.5° for PS85 and +1° for PS96, leaving the question of the observed 5° and 7° bias compared to radio soundings. To investigate this further we compared the winds measured on the crow's nest of the ship (König-Langlo, 2014b, 2016b). There are two anemometers (2D-sonic anemometers,

one at each side, König-Langlo et al. (2006)) mounted at a height of around 39 m above sea level. The first usable data points of the lidar measurements are at approximately 50 m height. Comparing the wind direction measured by the lidar in 50 m with wind direction in 60 to 200 m, we found an overall linear increase (decrease) of wind direction with height during PS85 (PS96). Assuming this change of wind direction is also present between the 39 m anemometer and the lidar data (approx. 50–75 m) this could lead to a slight positive (negative) bias during PS85 (PS96) of about 1°. An overview is shown in Figure 7 and the

statistics computed for this comparison are shown in Table 4.

Overall lidar and ship (anemometer) measurements agree well. In case of PS96 the comparison of the lidar to the ship anemometers suggests that the determined bias compared to radio soundings is also present. However, the anemometers are also disturbed by the ship's superstructure depending on wind direction (see section 2). One obvious explanation for the bias would be a misalignment of the lidar with respect to the ship. As an offset of 6° should be visible by eye and is not confirmed

by the analysis of the inclinometer correlation, the reason for the bias is still unclear.

## 4   Case Studies

In the following, we present two case studies. The first one focuses on the choice of the SNR threshold and the second one underlines the added value of lidar measurements compared to standard ship anemometer and radio sounding data.

### 4.1   PS85 - Arctic 2014/06/12

The beginning of the 12 June 2014 starts with wind speeds around 8.5 m s$^{-1}$ and wind from N-NW (Fig. 8) By midday, the wind decreases down to approx. 2 m s$^{-1}$ and the direction changes almost by 180°, thus from S-SW now. Weather charts for this day show that *Polarstern* was navigating through a synoptic high pressure ridge, which causes the measured wind changes.

The radiosonde wind profile at 1103 UTC agrees well with the lidar wind profiles at 1100 and 1109 UTC (Fig. 9), and the lidar data agree also with the ship wind measurements (Fig. 8). The potential temperature profile shows an almost neutral

stratification with high humidity topped by a strong inversion at 900 m. The plot for the SNR (Fig. 8, bottom) shows that with the conservative SNR threshold determined by the method presented in this study (17dB for PS85) the wind speed decreases in the afternoon would only be partially detected. However, this decrease below 250 m seems to be highly realistic in comparison



with the ship measurements. Extending the SNR threshold to -20 dB or -23 dB yields overall reasonable results, but adds also some noisy pixels particularly at the top height of the measurements. The presented method for determining a conservative SNR threshold seems to distinguish well between reliable and unreliable data. However, for specific cases it does make sense to check manually, if the limit can be extended to gain reliable data.

## 4.2   PS96 - Antarctic 2016/01/16 – 2016/01/17

The second case study is located in the Antarctic during PS96 (Fig. 10). It is chosen because it presents a situation of a stable boundary layer (SBL) with low-level jets (LLJs). The first LLJ was measured close after midnight at the 17 January 2016 between 0030 and 0230 UTC, and a second LLJ a few hours later between 0530 and 0730UTC, and the third LLJ between 1000 and 1130 UTC (Fig. 10, top). The LLJ wind speeds reached a maximum of up to 14 m s$^{-1}$ at a height of 200 m (Fig. 10, top). Three radio soundings are available at 16 January 1700 UTC, and for 17 January at 0700 and 1200 UTC. Only the profile at 0652 UTC on 17 January captured one of the LLJs (Fig. 11). The radiosonde profile agrees well with the lidar winds. The LLJ is located at the top of a surface inversion, and is associated with a strong directional shear in the lowest 200 m. It has to be noted that the ship was orientated perpendicular to the wind for this radiosonde launch, so that the ship's influence on the radiosonde winds was minimized for this LLJ situation. The short duration and fast developments of the LLJs illustrate the benefit of vertical wind profiles with high temporal resolution.

## 5   Conclusions

We presented a verification of wind speed profiles measured by a wind lidar during two cruises of the research vessel *Polarstern* in the Arctic and Antarctic. The lidar was not motion-stabilized, but ship motions and the ship's orientation were measured by the ship's navigation system and by a high-frequency Attitude Heading Reference System. The wind calculation is based on VAD scans with eight directions (rays), thus there is a high oversampling which allows for additional quality tests. We present a processing chain for the data, which includes a quality control for each ray and a method for deriving an empirical SNR threshold. This threshold can be calculated for individual measurements setups (e.g. different number of rays, averaging time), and robust thresholds of -17 dB and -20 dB are found for the Arctic and Antarctic cruise, respectively. Due to the oversampling, an error estimation of the lidar winds can be made, which can be used as additional quality criteria. The lidar wind profiles were compared with the routine meteorological measurements of the ship and radiosonde data. Overall, the radiosonde comparisons yield similar results as found in in Achtert et al. (2015) using as motion-stabilized lidar. The wind speed bias is very small (0.1 m s$^{-1}$) for our standard data processing and the RMSD is about 1 m s$^{-1}$. For wind direction, the RMSD is about 10°, but we also find a bias of 5°. In conclusion, the results of the postprocessing of non-motion stabilized lidar data achieve comparable good quality as the motion-stabilized lidar study of Achtert et al. (2015). The comparison with the routine wind measurements of the ship at 40 m height yields a larger data set and a similar bias and RMSD. It has also to be considered, that the wind field around the ship is influenced by the ship's superstructure, particularly if the ship is orientated into the wind. As this often occurs for radiosonde launches during the ship cruise, the lowest 50 m of the radiosonde wind profile should not be used for





these situations. Turning the wind perpendicular to the wind is desirable. The two case studies show that for special situations a flexible change of the SNR can be beneficial, and that ABL phenomena like short-lived LLJs are generally not captured by the routine radio soundings. The lidar with a high temporal resolution of 10–15 min can detect these phenomena, and would be ideally combined with a temperature profiler with a similar resolution. Alternatively, the lidar measurements can guide

dedicated radiosonde launches during future campaigns, since e.g. LLJs can be detected in real-time with the lidar.

## Appendix A

Given a measured Doppler velocities $d$ (negative if wind is blowing towards the lidar) in normalized directions $(x, y, z)$ (east, north, up) and the wind speed $(u, v, w)$ we have the following equation:

$$d = x \cdot u + y \cdot v + z \cdot w \tag{A1}$$

Transforming the wind $(u, v, w)$ to (ff = wind speed, dd = wind direction, $w$) with ff = dd = 0 if $u = v = 0$ we get

$$d = x \cdot \left( \cos(-\text{dd} - \frac{\pi}{2}) \cdot \text{ff} \right) + y \cdot \left( \sin(-\text{dd} - \frac{\pi}{2}) \cdot \text{ff} \right) + z \cdot w \tag{A2}$$

Transforming the direction $(x, y, z)$ to ($\theta$ = elevation angle, $\phi$ = azimuth angle starting north and turning clockwise) with $\phi = 0$ if $\theta = \pm 90° = \pm \frac{\pi}{2}$ we get

$$d = \left( \cos(-\phi + \frac{\pi}{2}) \cdot \cos(\theta) \right) \cdot \left( \cos(-\text{dd} - \frac{\pi}{2}) \cdot \text{ff} \right) + \left( \sin(-\phi + \frac{\pi}{2}) \cdot \cos(\theta) \right) \cdot \left( \sin(-\text{dd} - \frac{\pi}{2}) \cdot \text{ff} \right) + \sin(\theta) \cdot w \tag{A3}$$

Simplifying we get

$$d = \left( \cos(-\phi + \frac{\pi}{2}) \cdot \cos(-\text{dd} - \frac{\pi}{2}) + \sin(-\phi + \frac{\pi}{2}) \cdot \sin(-\text{dd} - \frac{\pi}{2}) \right) \cdot \text{ff} \cdot \cos(\theta) + \sin(\theta) \cdot w \tag{A4}$$

Using the trigonometric formula $\cos(a - b) = \cos(a) \cdot \cos(b) + \sin(a) \cdot \sin(b)$ we get

$$d = \left( \cos(-\phi + \frac{\pi}{2} + \text{dd} + \frac{\pi}{2}) \right) \cdot \text{ff} \cdot \cos(\theta) + \sin(\theta) \cdot w \tag{A5}$$

Simplifying we get

$$d = \cos(\phi - \text{dd} - \pi) \cdot \text{ff} \cdot \cos(\theta) + \sin(\theta) \cdot w \tag{A6}$$

*Competing interests.* The authors declare that they have no conflict of interest.

*Acknowledgements.* The measurements were performed during the two *Polarstern* cruises PS85 and PS96 funded by the Alfred-Wegner-Institute under Polarstern grants AWI_PS85_01 and AWI_PS96_03. The research was funded by the Federal Ministry of Education and





Research (Bundesministerium für Bildung und Forschung – BMBF) as part of the project 'Laptev-Sea Transdrift' under grant 03G0833D and by the SPP 1158 'Antarctic research' of the DFG (Deutsche Forschungsgemeinschaft) under grant HE 2740/19.

From all the software that was used we would like to highlight R, RStudio and the R-packages "doParallel" and "data.table". We thank our colleague Clemens Drüe for many useful discussions and our colleage Uwe Baltes for help with the cruise preparations.





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





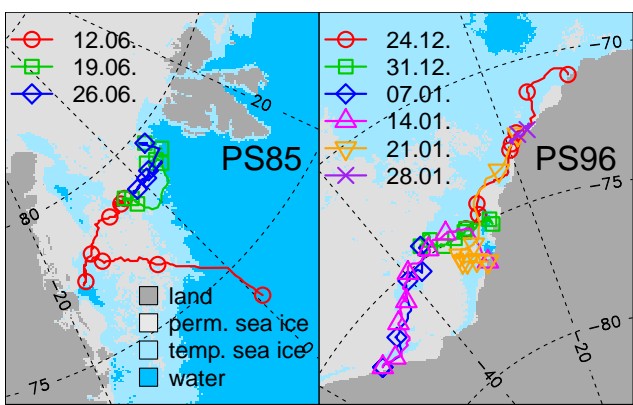

**Figure 1.** Cruise track of *Polarstern* during PS85 (left) and PS96 (right) with different colors for every week (symbol mark every day 0000 UTC). Beside land (dark gray) and water (dark blue) sea ice concentration (>15%) during the measuring period are shown: present every day (light gray) and present at least one day (light blue). Sea ice concentration taken from AMSR2 (Spreen et al., 2008)

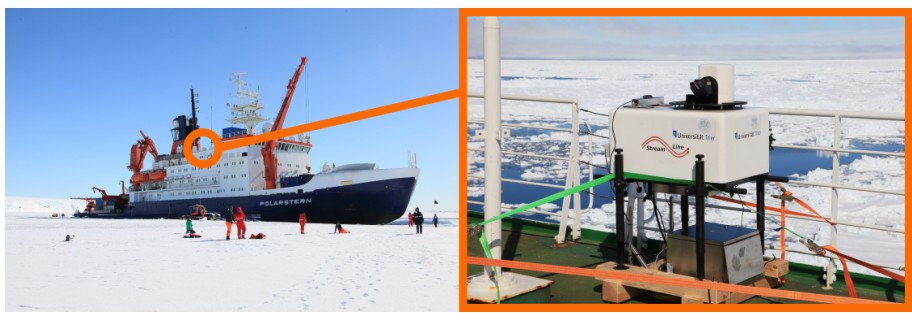

**Figure 2.** Position of the lidar on the RV *Polarstern*





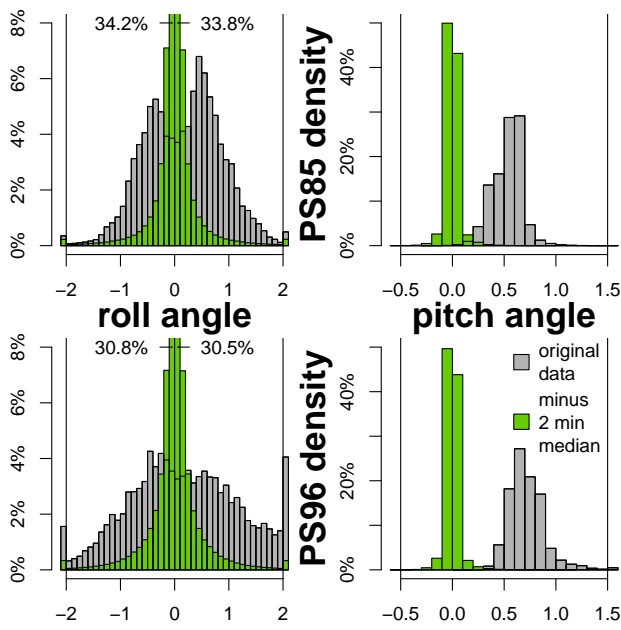

**Figure 3.** Frequency distribution of the ship angle (gray) and ship angle minus a 2-min running median (green) during the measurement time.

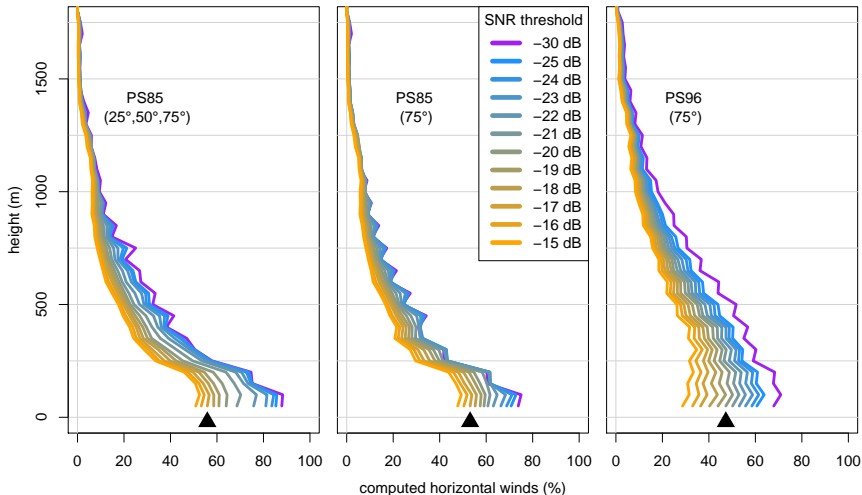

**Figure 4.** Percentage of wind calculations (speed/direction) from VAD scans as a function of SNR threshold at different heights during PS85 using all elevations (left), only elevation of 75° (middle) and PS96 (right). Overall number of VADs for PS85/PS96 was 3552/4250. The black triangle indicates the chosen SNR threshold based on Figure 5. For the height between 0 to 1750 m this chosen threshold results in 15% (PS85, all elevations) / 14% (PS85, only 75°) / 21% (PS96) computed horizontal winds.





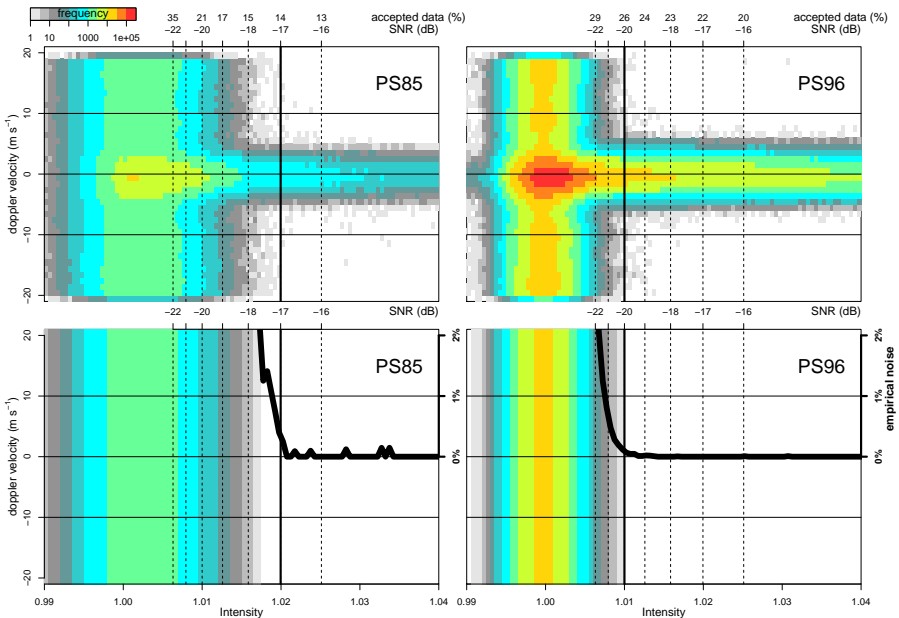

**Figure 5.** Top row: Frequency of Doppler velocities of VAD scans with 75° elevation depending on the intensity/SNR and for PS85 (left) and PS96 (right). Bottom row: Empirical noise computed as the mean for points above 10 m s$^{-1}$ or below -10 m s$^{-1}$. The solid black line shows the ratio of empirical noise and all measured data (top) at each intensity/SNR. On the top axis it is also noted how much data would be accepted if the respective (minimal) SNR would be chosen.

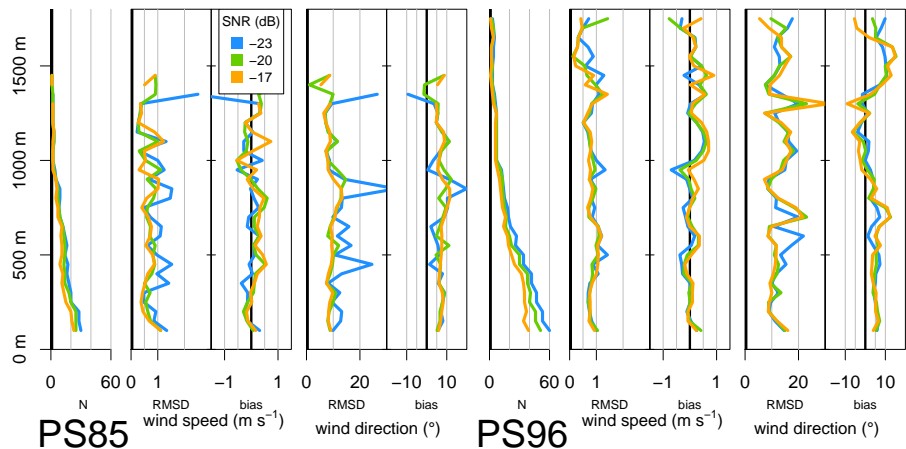

**Figure 6.** RMSD, bias and number of used radio soundings (N) by height of wind speed and direction for PS85 (left) and PS96 (right). Different colors show different SNR thresholds (-23 dB blue, -20 dB green, -17 dB orange). Only scans with an elevation of 75° were used.





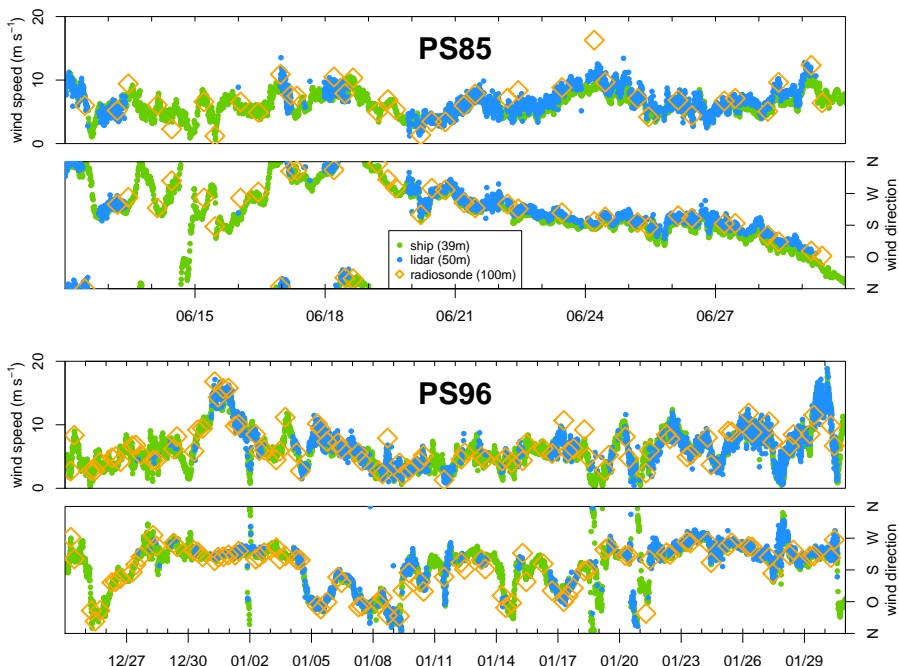

**Figure 7.** Comparison of wind speed (ff) and wind direction (dd) between lidar at 50 m height (blue) and ship anemometer (green) for PS85 (top) and PS96 (bottom). Radiosonde winds at 100 m are marked (orange diamond) for reference. Only scans with an elevation of 75° were used.





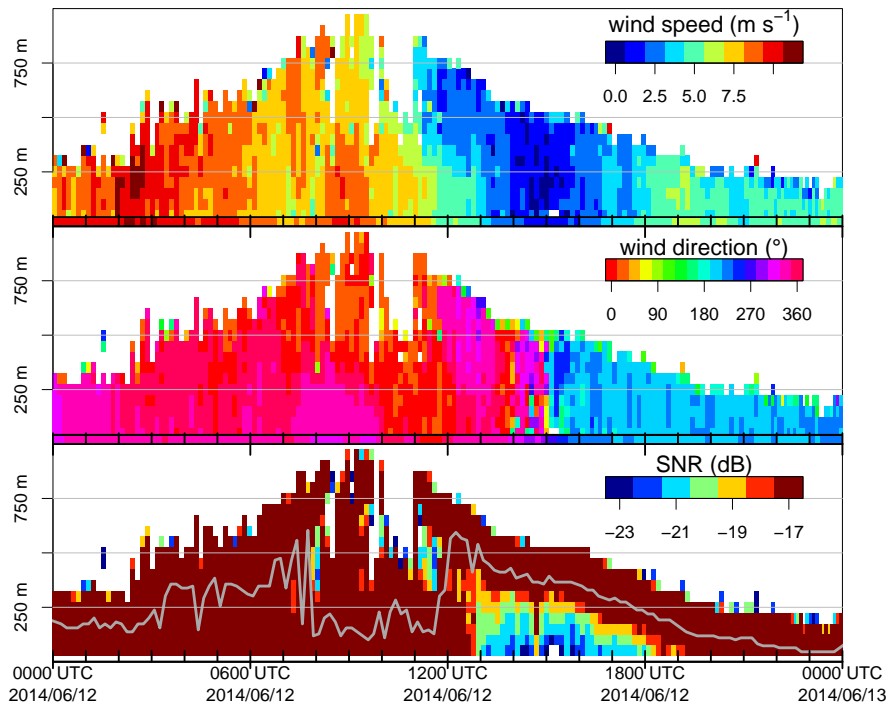

**Figure 8.** Lidar wind speed (top) and direction (middle) for -23 dB SNR threshold for the 12 June 2014 (location see PS85 in Fig. 1). Colors below the black line (40 m) show the wind measurements of RV *Polarstern* (anemometer). The bottom plot presents the SNR thresholds that would allow for a wind calculation. Grey line is the cloud base from ceilometer measurements of RV *Polarstern*.

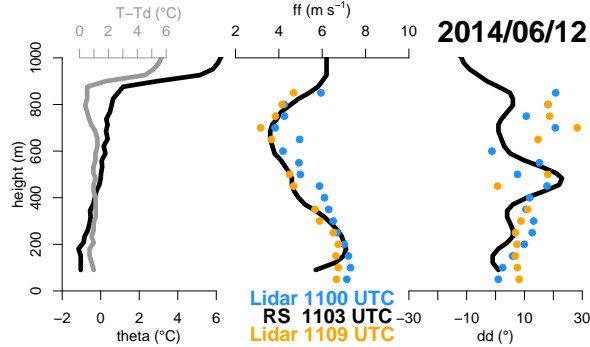

**Figure 9.** Vertical profiles of potential temperature (theta), dew-point spread (T-Td), wind speed (ff) and direction (dd) of radiosondes vs lidar wind speed and direction for around 1100 UTC 12 June 2016. A SNR threshold of -23 dB was used.





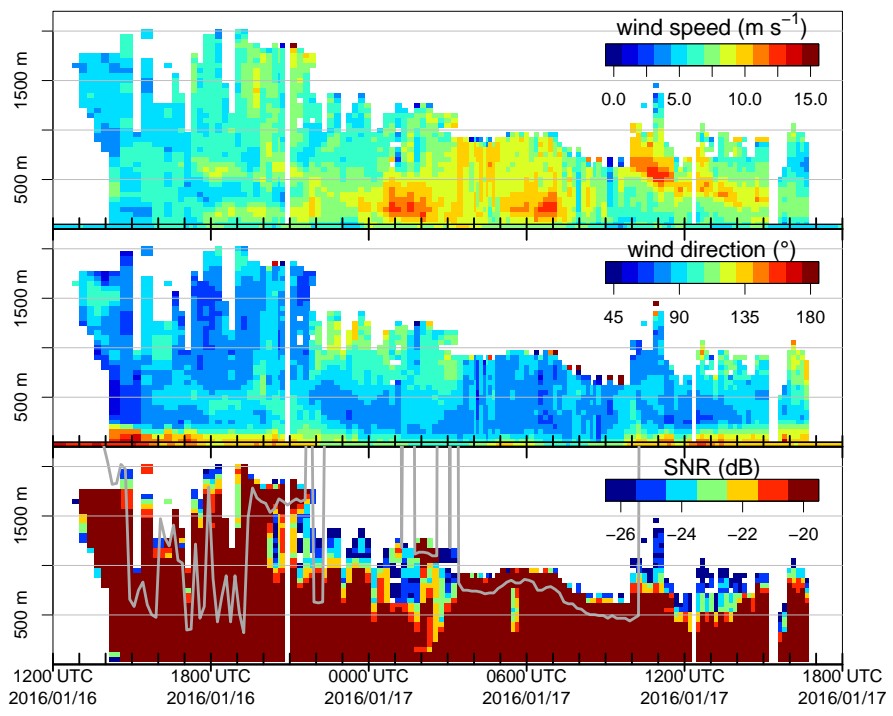

**Figure 10.** As Figure 8 but for 16 and 17 January 2016 (Antarctic, PS96) and with a -26 dB SNR threshold.

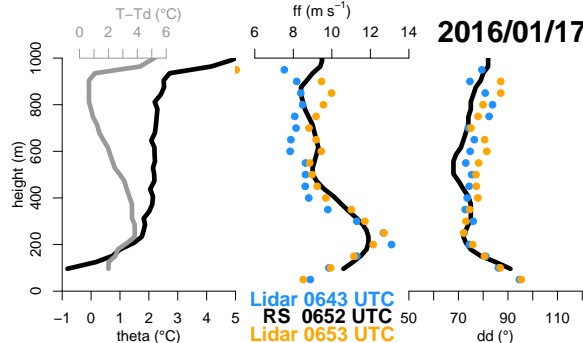

**Figure 11.** As Figure 9 but for the LLJ around 0700 UTC 17 January 2016 (PS96). A SNR threshold of -26 dB was used.





**Table 1.** Characteristics of the lidar measurements.

| | |
|---|---|
| wavelength | 1.5 $\mu$m (eye-safe, class 1m) |
| Gate length | 18 m |
| Points per gate | 6 (overlapping for PS96) |
| Band width | $\pm$19.4 m s$^{-1}$ |
| Resolution | 0.038 m s$^{-1}$ |
| Threshold for signal-to-noise ratio (SNR) | variable (default -20 dB) |
| Measurement error | ca. 0.1 m s$^{-1}$ (depending on SNR) |
| Pulse rate | 10 kHz |
| Beam range | 30–3600 m |
| Beam focus | variable (300–1800 m) |
| Averaging time | variable (1–30 s) |
| Scanning horizontal | 0° to 360° |
| Scanning vertical | -15° to 90° |

**Table 2.** Statistics for all available lidar data compared to radio soundings. M indicates the number of used radio soundings. N indicates the number of compared measurements (N is lower for the wind direction because up to six cases with wind speed < 0.5 m s$^{-1}$ are removed). PS85 computed for -17 dB SNR threshold with only 75° elevation scans (first column, as shown in Fig. 6) and with all 25°, 50° and 75° (second column). PS96 computed for 20 dB SNR threshold with default case (standard deviation of yaw angle below 2° for each ray; third column, as shown in Fig. 6) and a stricter case (standard deviation of yaw angle below 0.5° for each ray; fourth column). aR is the correlation coefficient for angular variables.

| | | | wind speed in m s$^{-1}$ | | | wind direction in deg | | |
|---|---|---|---|---|---|---|---|---|
| | M | N | RMSD | bias | $R^2$ | RMSD | bias | aR$^2$ |
| PS85 (VAD with 75°) | 28 | 216 | 0.7 | 0.1 | 0.95 | 9 | 7 | 0.99 |
| PS85 (25,50,75°) | 28 | 227 | 0.7 | 0.0 | 0.95 | 6 | 3 | 0.99 |
| PS96 (2° yaw-sd) | 58 | 574 | 0.9 | 0.1 | 0.95 | 13 | 5 | 0.96 |
| PS96 (0.5° yaw-sd) | 49 | 502 | 0.8 | 0.0 | 0.96 | 13 | 5 | 0.95 |





**Table 3.** Statistics as in Table 2, but showing the range of the statistic variables for different computations. These includes all possible combination of the following two (default marked with *): (1) the thickness of layers and thus the interpolation in height of lidar data [10, 20, 30, 40, 50* m] and (2) the time range of used lidar measurements around the radio sounding measurement (100 s after start) [± 5, 10*, 15, 30 min].

| | M | N | wind speed in m s$^{-1}$ | | | wind direction in deg | | |
| --- | --- | --- | --- | --- | --- | --- | --- | --- |
| | | | RMSD | bias | R$^2$ | RMSD | bias | aR$^2$ |
| PS85 (VAD with 75°) | 27 − 28 | 192 − 489 | 0.7 | 0.1 − 0.2 | 0.94 − 0.96 | 8 − 9 | 7 − 8 | 0.99 |
| PS85 (25,50,75°) | 28 | 209 − 508 | 0.6 − 0.7 | -0.1 − 0.0 | 0.95 − 0.97 | 6 | 3 | 0.99 |
| PS96 (2° yaw-sd) | 39 − 60 | 368 − 1391 | 0.7 − 0.9 | 0.0 − 0.1 | 0.95 − 0.96 | 13 − 16 | 5 − 6 | 0.95 − 0.97 |
| PS96 (0.5° yaw-sd) | 32 − 51 | 315 − 1226 | 0.7 − 0.8 | 0.0 − 0.1 | 0.96 | 13 − 17 | 5 − 6 | 0.94 − 0.96 |

**Table 4.** Statistics for computed lidar data points compared ship anemometer (39 m); standard case 50 m (representing approx. 50–75 m); N indicates the number of compared measurements (N is lower for the wind direction because up to 1 case with wind speed < 0.5 m s$^{-1}$ is removed). aR is the correlation coefficient for angular variables.

| | N | wind speed in m s$^{-1}$ | | | wind direction in deg | | |
| --- | --- | --- | --- | --- | --- | --- | --- |
| | | RMSD | bias | R$^2$ | RMSD | bias | aR$^2$ |
| PS85 (VAD with 75°) | 1886 | 1.1 | 0.5 | 0.76 | 19 | 15 | 0.95 |
| PS85 (25,50,75°) | 1984 | 0.6 | 0.0 | 0.87 | 11 | 7 | 0.98 |
| PS96 | 2010 | 1.0 | 0.0 | 0.93 | 14 | 4 | 0.94 |