# Peer review of "Analysis of the performance of a ship-borne scanning wind lidar in the Arctic and Antarctic"

_Atmospheric Measurement Techniques, 2018_

## Referee Comment (RC1) · Anonymous Referee #1 · 18 Jul 2018

The paper by Zentek et al. describes the use of a scanning lidar for ship-borne wind measurements without a motion stabilisation platform. The authors used data collected with an external Altitude Heading Reference System to correct for the ship's pitch and roll after the measurement campaign. The presented technique and the statistical comparison of the lidar wind measurements to radio soundings as well as to ship measurements is important for the scientific community due to the clear need for wind measurements over the oceans – especially in the polar regions. Such measurements are important for a better understanding of atmospheric processes in the maritime environment. The paper is suitable for publication in AMT and can be published after minor revision.

Major comment:

[Figure]

- The lidar measurements that have been corrected for the ship's pitch and roll after the measurements are performed consist of profiles that are the average of 12 to 15 seconds of individual rays for the PS96 campaign and 1.5 seconds for the PS85 campaign. The movement of the ship during these averaging periods introduces horizontal wind components into the vertical wind. This is an important source of error and should be discussed in the paper. How does the proposed methodology account for movements during the time needed to obtain the averaged profiles that are later motion corrected?

Other comments:

- Page 5, line 12: Can you really assume horizontal homogeneous wind fields? The elevation changes during the scan.

- Page 6 line 30: Doppler velocity due to horizontal wind speed is less than 26 % at this elevation... Is that still true if you correct pitch and roll after the measurements were taken? Your elevation is not stable at 75° due to the ship's motion.

- Page 7 line 9-10: What are the reasons for the different SNR thresholds for the two campaigns? Could it be the different averaging times of the rays? The elevation is not stable during the measurements and you get different horizontal wind components into your vertical wind component. With a longer averaging time the effect might be enhanced.

- Table 2 and Table 3: Similar to previous comment the statistics for the PS85 campaign with a shorter averaging time are better than for PS96 with a longer averaging time. What is the reason for this?

- Page 8 line 19/20: Could the higher bias be explained by not having a horizontally homogeneous wind field? You only correct for the elevation and azimuth but you cannot correct for the horizontal wind component being present in the vertical wind component.

- Figure 6: please add a plot for the relative difference between lidar and radio soundings by height for wind speed and wind direction.

- Figure 7: Please add a plot for relative difference for the comparison of wind speed and wind direction for lidar and radio soundings as well as for lidar and ship anemometer.

- Figure 9: It looks like the lower SNR values between 300 and 600 m Figure 8 (bottom) have more influence on the wind direction than the wind speed. What would be the reason?

[Figure]

---

## Referee Comment (RC2) · Anonymous Referee #2 · 3 Aug 2018

The manuscript (Zentek et al.) explains how a commercial Doppler lidar (HALO Streamline) is operated on RV Polarstern. The results are compared to standard measurements of radiosondes and sonic anemometers onboard. The lidar was operated during two campaigns in the Arctic and Antarctic. Such measurements in the changing Arctic regions specifically and on the sea, in general, are of importance as those places lack such data. The manuscript focuses on the technical aspects of operating the HALO and analyzing the datasets from the ship. The wind profiles measured from the Doppler lidar agree well with the other sensors, as shown in many other studies before. Although the steps taken to derive wind profiles are fine the authors should write more explicitly what is new (approaches or findings) compared to other similar measurements. One general statement of the manuscript seems to be that the ac-

tive stabilization of the Doppler lidar is not required as shown as in Achtert et al. It should be explicitly stated in the conclusions that this is probably true only for measurements of horizontal winds with the VAD technique. Measurements in PPI scanning mode configurations, or even more importantly turbulence and sedimentation-speed measurements of clouds and ABL vertical-wind measurements are very strongly influenced by the motion of the ship. I found it a bit confusing to see three different things called "NOISE" in the manuscript. There is the signal detection noise in the SNR (which also determines the SNR threshold), then there is the error of the line-of-sight wind estimator (peak-finding accuracy or similar, which is connected to the Cramer-Rao Lower Bound theorem), there are outliers if the wind estimator fails, and finally there is the error of the VAD result on the final horizontal wind by non-perfect compensation of the ship's attitude. These quantities should be differentiated more carefully in the manuscript.

Minor comments: -The abstract is too long and can be shortened. It shouldn't include a motivation and lengthy formulations. Just the facts in a very condensed form. -RMSD is not explained -P1L19: the abbreviation AOI is not really needed as it is never used again in the manuscript. (It could be mentioned that AOI interactions are strongly related to turbulent processes in the ABL which can be observed with a lidar, too. Even though turbulence parameters and not measured here.) -P2: Most of the introduction/literature review deals with the specifics of the HALO lidar and not for Doppler lidars in general. This should be mentioned or revised. -P3L21: Could you please describe the HALO configuration in more detail? How can there be a 3m range resolution? I assume that the laser pulse is much longer? The effect of overlapping gates and non-independent measurements at those range gates should be mentioned. A bit is seen in Fig. 4, but the explanation could be more specific for the zigzag lines. -P4L34: ..."if data quality is not of importance". Better mention the errors with and without correction. This could help a reader to evaluate the effects of a/no stabilization better. -P5L6: How it the SNR defined for the HALO? This is important to follow the upcoming discussion about the thresholds. Some people also derive the CNR to be

more correct. I would like to see a bit more discrimination between those terms here or at P6L17. -P5L20: Could you please find formula signs (one character) for wind speed and direction other than dd and ff? -P5L25: 1st: "has a fixed elevation ANGLE". And 2nd: Is that really true for a ship? -P6L17: Should you even expect that the SNR threshold is the same for every of the HALO instrument? I'm not sure that these thresholds can be really compared. Maybe the laser power/pulse length/DAQ bit resolution is different? Again, it depends, how the dB's are defined here. -P6L22: Is it really true, that the spectra or ACF cannot be stored? One should always try to store the spectra (at least for a while) in low-signal regimes like the Arctic so that later post-averaging is possible to increase the SNR. -P6L26-27: If the PS96 data contain less noise then I would expect that the SNR is higher. How can this mean, that you need a different SNR threshold? This is a bit counterintuitive. Except the so-called "SNR" is the signal and not an actual SNR? -P6L27-30: Please explain why the vertical wind is around 0. This depends on the averaging period, on the precision of the angle of vertical stare (especially on a ship), and sometimes even on the synoptic situation (stationary waves, etc). In fact, you can determine the noise of your LOS wind by looking at the difference between the Autocovariance function at 0 and at the first leg. Or by evaluating the high-frequency tail of the wind power spectrum. And this could be done for different SNR thresholds. -P7L1-12: this paragraph is a bit hard to understand. Can you explain why you do not use the goodness of the VAD fit to determine when a VAD delivered good and bad results? -P7L5: It is mentioned that the VAD results are averaged for 20 min. But since it is possible that there are outliers in those VAD results shouldn't the median be a better indicator here? -P7-8L20: The bias of 10° for the wind direction is quite unexpected. One would assume that radiosondes and the lidar both use GPS? Or is there a magnetic compass involved somewhere which might show a bias in Polar regions? It is a bit unsatisfactory that the reason for this bias remains unclear here. What is the VAISALA specification for their wind-direction bias? -P8L31: here it should be -17dB. -P9L5-15: The occurrence of these 3 fast LLJs is interesting. Can you give any explanation of the processes? Usually, in these latitudes, the typical Ekman oscillation time should be 12 hrs. So there must be another cause for these LLJs. Maybe you could reference some work on Arctic LLJ (e.g. Jacobson, ACP, 2013). -P9L17: A bit general comment: I believe the topic of the paper should not be: The radiosondes are correct and let's see how well the lidar can be verified by this. A Doppler lidar by its design is one of the best methods to measure wind. In the end, it just depends on how precise one can measure frequencies. It is more a question of how much errors are generated by a moving platform like a ship. And what are the best scan strategies on a ship? It would be nice to get some answer to these questions, as other researchers might profit from this. -P9L27: "The RMSD is 10° but we also find 5°". This sentence needs to be revised. -P10L1: "turning the wind perpendicular to the wind". Something is wrong here.

-Fig.3: Can this also be done for the YAW angle? It seems the wind direction error bias is highest. So probably one should have a look at this angle, too. -Fig.5: If Intensity is SNR+1, why does the axis start at 0.99? -Fig.7: It would be nice to include a correlation plot. -Fig.8: What is seen on the y-axis? Height? If it is height, how can there be measurements several hundred meters above the cloud base? Please explain in the text. -Fig.9: Theta can be omitted and the x-axis can be annotated "POT. TEMPERATURE (°C)". Also, the date can be omitted since it is mentioned in the figure caption. -Table 1: The SNR threshold is -20dB, but in the text, it is written -18.2dB is from the manufacturer. So what means "default"?

---

## Author Comment (AC1) · 27 Aug 2018

*COMMENT [RC1]: The lidar measurements that have been corrected for the ship's pitch and roll after the measurements are performed consist of profiles that are the average of 12 to 15 seconds of individual rays for the PS96 campaign and 1.5 seconds for the PS85 campaign. The movement of the ship during these averaging periods introduces horizontal wind components into the vertical wind. This is an important source of error and should be discussed in the paper. How does the proposed methodology account for movements during the time needed to obtain the averaged profiles that are later motion corrected?*

We address this issue in section 2.3.1 "Ship motion correction" (page 4, line 23ff Fig.3)

[Figure]

but never stated clearly what we address and just refer to it as "the error" (page 4, line 25). We corrected this

Before:

During PS96 the averaging time of a single ray was typically 12–15 seconds, so that we corrected each single measurement with the mean value over the averaging time. All measurements that have a standard deviation of roll or pitch angle larger than 0.5° or yaw angle larger than 2° over this averaging time were excluded from the analysis in order to reduce the error. [... other text ...] For a data point in 1 km distance from the lidar a change of elevation from 75° to 75.5° (25° to 25.5°) causes a difference in height of 2 m (8 m) and a horizontal wind speed error of less than 3.3% (0.4%). This is acceptable as we will later interpolate over height intervals of 50 m.

Now:

During PS96 the averaging time of a single ray was typically 12–15 seconds, so that we corrected each single measurement with the mean value over the averaging time. **This introduces an error whenever the ship's angle and thus the lidar angle changes during this averaging time. In order to reduce the error,** all measurements that have a standard deviation of roll or pitch angle larger than 0.5° or yaw angle larger than 2° over this averaging time were excluded from the analysis. **Correcting the direction of the lidar measurement by the mean roll and pitch angle during the averaging time should already cause most of the error to average out, as it measures partly too much and partly too less wind speed. But even if this is not the case, for** a data point in 1 km distance from the lidar a change of elevation from 75° to 75.5° (25° to 25.5°) causes a difference in height of 2 m (8 m) and **the resulting** horizontal wind speed error **is** less than 3.3% (0.4%). This is acceptable as we will later interpolate over height intervals of 50 m **and only evaluate the horizontal wind in our paper**.

*COMMENT [RC1]: Page 5, line 12: Can you really assume horizontal homoge-*

**neous wind fields? The elevation changes during the scan.**

We assume the homogeneity for a fixed height and only take data points from that height (page 5, line 10f). A different elevation as such is no problem. For example the wind velocity could still be computed if there are 3 different measurements with elevations of 50, 60 and 70$^\circ$ and azimuths of 20, 30, and 40$^\circ$. The error resulting from the change of elevation (due to the ships movement) during each single ray was discussed in section 2.3.1.

**COMMENT [RC1]: Page 6 line 30: Doppler velocity due to horizontal wind speed is less than 26 % at this elevation. Is that still true if you correct pitch and roll after the measurements were taken? Your elevation is not stable at 75$^\circ$ due to the ship's motion.**

Yes, a Doppler velocity of 10 $ms^{-1}$ for an elevation of 73/75/77$^\circ$ would result in 34/39/44 $ms^{-1}$ horizontal wind speed. These values can be safely considered to be unrealistic for our conditions.

**COMMENT [RC1]: Page 7 line 9-10: What are the reasons for the different SNR thresholds for the two campaigns? Could it be the different averaging times of the rays? The elevation is not stable during the measurements and you get different horizontal wind components into your vertical wind component. With a longer averaging time the effect might be enhanced.**

The reviewer is partially correct. As stated on page 6 line 15, the value for a SNR threshold can vary depending on the instrument specific performance (detector noise) and the variability of atmospheric conditions within the measured volume. We think that the main reason is not the influence of elevation but the averaging time itself. We changed the passage (page 6, line 25-27) explaining this.

Before:

Additionally due to the different averaging time for each ray during PS85 and PS96 (1.5 vs 12–15 sec), the PS96 data contains less noise and thus it makes sense to choose a different SNR threshold for each data set.

After:

Additionally due to the longer averaging time for each ray during PS96 (12–15 sec) than during PS85 (1.5 sec), the PS96 data **allow for a lower SNR threshold compared to the PS85 data, because averaging over a longer period given the same SNR results in better data.** Thus, it makes sense to choose a **less strict** SNR threshold for the PS96 data **set to make both data sets more comparable**.

*COMMENT [RC1]: Table 2 and Table 3: Similar to previous comment the statistics for the PS85 campaign with a shorter averaging time are better than for PS96 with a longer averaging time. What is the reason for this?*

We think our data sets are too small to reach any definite conclusion or even to be reasonably certain that one data set / scanning technique is really better than the other. We think it is possible, that external sources like a bias in weather condition during one cruise could be reason enough to cause this differences. One reason to include Table 3 in this paper was to show that a different reasonable analysis-configuration would lead to different statistics. For example the computed RMSD for PS96 can change from 0.9 to 0.7 $ms^{-1}$ making it the same as for PS85.

*COMMENT [RC1]: Page 8 line 19/20: Could the higher bias be explained by not having a horizontally homogeneous wind field? You only correct for the elevation and azimuth but you cannot correct for the horizontal wind component being present in the vertical wind component.*

We see no reason why this would lead to a positive bias in wind direction for both cruises and for both comparisons (radio sounding and anemometer).

**COMMENT [RC1]: Figure 6: please add a plot for the relative difference between lidar and radio soundings by height for wind speed and wind direction.**

We do not know of any definition for relative differences of wind directions. (Just labeling the axis differently by dividing by 180°?)

We plotted the relative difference for wind speed and absolute difference for wind direction with the plus symbol for each single case. We also plotted the mean relative difference for wind speed as lines scaled with a factor of 4. (Appended Fig. 1 and 2)

We do not think that the benefit is high enough to add this additional plots to the paper (But we change the Figure 6 by adding a little space between the RMSD and bias subplots.)

**COMMENT [RC1]: Figure 7: Please add a plot for relative difference for the comparison of wind speed and wind direction for lidar and radio soundings as well as for lidar and ship anemometer.**

We added the relative differences for wind speed and absolute difference for wind direction. We split the old figure into two separate once. (Appended Fig. 3 and 4)

**COMMENT [RC1]: Figure 9: It looks like the lower SNR values between 300 and 600 m Figure 8 (bottom) have more influence on the wind direction than the wind speed. What would be the reason?**

The higher scatter of lidar wind directions between 400 and 700 m in Fig.9 are more likely associated with lower wind speeds.

[Figure]

[Figure]

**Fig. 1.** Figure 6 with relativ differences for PS85

[Figure]

**Fig. 2.** Figure 6 with relativ differences for PS96

**PS85**

Legend:
- ship (39 m)
- lidar (50 m)
- radiosonde (100 m)

**Fig. 3.** Figure 7 with relativ differences for PS85

**PS96**

legend:
- ship (39 m)
- lidar (50 m)
- radiosonde (100 m)

**Fig. 4.** Figure 7 with relativ differences for PS96

---

## Author Comment (AC2) · 27 Aug 2018

*COMMENT [RC2]: Although the steps taken to derive wind profiles are fine the authors should write more explicitly what is new (approaches or findings) compared to other similar measurements.*

We think that is already mentioned in the abstract: 1) first ship-based lidar measurements in the Antarctic, 2) assessing the quality of wind profiles of a non-motion stabilized Doppler lidar operated on an icebreaker, 3) empirical SNR method. We repeat this information now in the conclusion section in the following way:

...high-frequency Attitude Heading Reference System. **This is the first time that a wind lidar was operated on an icebreaker in the Antarctic. A processing chain**

[Figure]

including quality control tests with a new empirical SNR threshold method and an error quantification is presented.

*COMMENT [RC2]: One general statement of the manuscript seems to be that the active stabilization of the Doppler lidar is not required as shown as in Achtert et al. It should be explicitly stated in the conclusions that this is probably true only for measurements of horizontal winds with the VAD technique. Measurements in PPI scanning mode configurations, or even more importantly turbulence and sedimentation-speed measurements of clouds and ABL vertical-wind measurements are very strongly influenced by the motion of the ship.*

We added this information in the conclusion

Before:

In conclusion, the results of the postprocessing of non-motion stabilized lidar data achieve comparable good quality as the motion-stabilized lidar study of Achtert et al. (2015).

Now:

In conclusion, the results of the postprocessing of non-motion stabilized lidar data achieve comparable good quality as the motion-stabilized lidar study of Achtert et al. (2015). **As our study focuses only on horizontal winds it should be noted that the influence on vertical wind and turbulence measurements is higher and was not evaluated. The need of a motion stabilized lidar for those measurements could be very important.**

*COMMENT [RC2]: I found it a bit confusing to see three different things called "NOISE" in the manuscript. There is the signal detection noise in the SNR (which also determines the SNR threshold), then there is the error of the line-of-sight wind estimator (peak-finding accuracy or similar, which is connected to*

*the Cramer-Rao Lower Bound theorem), there are outliers if the wind estimator fails, and finally there is the error of the VAD result on the final horizontal wind by non-perfect compensation of the ship's attitude. These quantities should be differentiated more carefully in the manuscript.*

We agree and tried our best to distinguish between error, SNR and empirical noise. Apart from our revisions due to the minor comments from the reviewers (e.g. P6L26-27) we searched the paper for the words "noise", "SNR" and "error" and made the following changes:

Page 1 line 13 - Change: "SNR" to "SNR threshold"

Page 5 line 6

Before:

First a signal-to-noise ratio (SNR) threshold was chosen and all data points within one ray with a worse SNR were removed.

Now:

First a signal-to-noise ratio (SNR) threshold was chosen and all data points within one ray with a worse SNR were removed. **The SNR is a value given in the lidar output for each scanned Doppler velocity value. It is separate from the empirical noise defined in section 2.3.3 as well as from the "noisy influence" due to other error sources like uncertainties related to the ships movement.**

Page 7 line 1

Before:

Data points outside this range can be regarded as noise. This condition is used to find a SNR threshold in a three-step procedure. First, we look at the overall frequency distribution of measured Doppler velocities (Fig. 5, top). We assume that the data mainly consists of two parts: the noise (homogenous along all wind speeds; top to

bottom) and the wind signal (relatively homogenous along the signal intensity or SNR; left to right). Signal intensity is defined as SNR+1. All points above 10 $ms^{-1}$ or below -10 $ms^{-1}$ are taken to construct a noise distribution as a function of intensity using the mean value (Fig. 5, bottom). We call this the empirical noise. We call this the empirical noise. In the second step, we take the ratio of the empirical noise and the mean of the measured Doppler velocities for each intensity, which results in an empirical noise fraction (plotted as solid line in Fig. 5, bottom). The noise fraction is close to zero for high intensities and starts to increase rapidly at different SNR values for both data sets.

Now:

Data points outside this range can be regarded as **wrong (or empirical noise)**. This condition is used to find a SNR threshold in a three-step procedure. First, we look at the overall frequency distribution of measured Doppler velocities (Fig. 5, top). We assume that the data mainly consists of two parts: the **empirical** noise (homogenous along all wind speeds; top to bottom) and the wind signal (relatively homogenous along the signal intensity or SNR; left to right). Signal intensity is defined as SNR+1. All points above 10 $ms^{-1}$ or below -10 $ms^{-1}$ are taken to construct **an empirical** noise distribution as a function of intensity using the mean value (Fig. 5, bottom). In the second step, we take the ratio of the empirical noise and the mean of the measured Doppler velocities for each intensity, which results in an empirical noise fraction (plotted as solid line in Fig. 5, bottom). The **empirical** noise fraction is close to zero for high intensities and starts to increase rapidly at different SNR values for both data sets.

Page 9 line 2 - Change: "noisy pixels" to "outliers"

Page 10 line 2 - Change: "SNR" to "SNR threshold"

Fig 8 and 10 - Change: "SNR" to "SNR threshold"

***COMMENT [RC2]: The abstract is too long and can be shortened. It shouldn't in-clude a motivation and lengthy formulations. Just the facts in a very condensed***

*form.*

We removed the first two (motivating) sentences of the abstract, but think that the other formulations are adequate.

Removed lines from abstract:

**Profiles of wind speed and direction at high spatial and temporal resolution are fundamental meteorological quantities for studies of the atmospheric boundary layer. Ship-based Doppler lidar measurements can contribute to fill the data gap over oceans particularly in polar regions.**

*COMMENT [RC2]: RMSD is not explained*

Changed in the abstract from "RMSD" to "root-mean-square deviation" Added in with the first appearance in the text "root-mean-square deviation (RMSD)"

*COMMENT [RC2]: P1L19: the abbreviation AOI is not really needed as it is never used again in the manuscript. (It could be mentioned that AOI interactions are strongly related to turbulent processes in the ABL which can be observed with a lidar, too. Even though turbulence parameters and not measured here.)*

We changed the two occurrences of "AOI" to "atmosphere-ocean-ice".

*COMMENT [RC2]: P2: Most of the introduction/literature review deals with the specifics of the HALO lidar and not for Doppler lidars in general. This should be mentioned or revised.*

This is now mentioned.

Before:

In synergy with additional remote sensing instruments measuring the temperature profile, the turbulent mixing conditions in the ABL can be described at high temporal and

vertical resolution of 10 min and 10 m, respectively (Brooks et al., 2017).

Now:

In synergy with additional remote sensing instruments measuring the temperature profile, the turbulent mixing conditions in the ABL can be described at high temporal and vertical resolution of 10 min and 10 m, respectively (Brooks et al., 2017). **Note that our literature research was focus on lidars similar to our own, thus it is likely biased towards lidars from the same manufacturer.**

*COMMENT [RC2]: P3L21: Could you please describe the HALO configuration in more detail? How can there be a 3m range resolution? I assume that the laser pulse is much longer? The effect of overlapping gates and non-independent measurements at those range gates should be mentioned. A bit is seen in Fig. 4, but the explanation could be more specific for the zigzag lines*

Our usage of the term "gate" was misleading/inconsistent. The HALO lets one choose the gate length by selecting in multiples of 6 m. In the software you choose 2*n points per gate, where each point increases the range length by 3 m. We changed our phrasing making it also consistent with Table 1.

Before:

One ray is divided into gates of 3 m length and the measured Doppler velocity is representative for six gates (18 m). During PS85 those six gates were non-overlapping, thus measurements were available every 18 m. During PS96 the six gates were overlapping, thus measurements were available every 3 m.

Now:

One ray is divided into **sections** of 3 m length and **one** measured Doppler velocity is representative for **gate length of** six **sections** (18 m). During PS85 those six **sections** were non-overlapping, thus measurements were available every 18 m. During PS96

the six **sections** were overlapping, thus measurements were available every 3 m. **But the measurements with overlapping sections are not independent as they are computed based on partially same data.**

*COMMENT [RC2]: P4L34: ..."if data quality is not of importance". Better mention the errors with and without correction. This could help a reader to evaluate the effects of a/no stabilization better.*

We said: "**high** data quality". We think this is already given in the sentence following "For example, [. . .] causes [. . .] wind speed error of less than 13% [. . .]"

*COMMENT [RC2]: P5L6: How it the SNR defined for the HALO? This is important to follow the upcoming discussion about the thresholds. Some people also derive the CNR to be more correct. I would like to see a bit more discrimination between those terms here or at P6L17.*

We would like to stay with SNR, which is used in many other studies. We added the following sentences near page 6 line 17 to explain how the SNR is computed:

**The background noise is usually measured at least once a day and at most every hour. For this, the scanning head is turned away from the sky towards the lidar casing and measures the signal while sending no pulses out. Thus the background noise can vary with time and operating conditions and can be different for different HALO instruments. To compute the SNR this signal strength of the background noise is subtracted from the signal strength of the measurement and afterwards divided by the signal strength of the background noise. If the signal during a measurement is lower than during the background noise scan, it can therefore cause a negative SNR. In general, more background noise scans were performed during PS85, but we didn't investigate the background noise further.**

*COMMENT [RC2]: P5L20: Could you please find formula signs (one character)*

*for wind speed and direction other than dd and ff?*

We changed it to "$v_h = sqrt(u^2 + v^2)$" and "$phi_h$" for wind direction

*COMMENT [RC2]: P5L25: 1st: "has a fixed elevation ANGLE" And 2nd: Is that really true for a ship?*

In our setup not. But with a stabilizing platform it would be.

Before:

Assuming that the lidar remains stationary and has a fixed elevation $\theta$, the equation further simplifies to

Now:

Assuming that the lidar remains stationary and has a fixed elevation **angle** $\theta$ **(which is not the case in our setup)**, the equation further simplifies to

*COMMENT [RC2]: P6L17: Should you even expect that the SNR threshold is the same for every of the HALO instrument? I'm not sure that these thresholds can be really compared. Maybe the laser power/pulse length/DAQ bit resolution is different? Again, it depends, how the dB's are defined here.*

Yes comparability is a big issue. As we show in section 2.3.3 with Fig.5, even the same lidar with another averaging time demands a different SNR threshold. Also we do not know of any common definition of a SNR threshold so a comparison is already problematic here. We just gave a short overview of values in literature and assumed that the common definition of dB in the context of a lidar is $10log_{10}(x)$.

*COMMENT [RC2]: P6L22: Is it really true, that the spectra or ACF cannot be stored? One should always try to store the spectra (at least for a while) in low-*

*signal regimes like the Arctic so that later post-averaging is possible to increase the SNR.*

Correction / Clarification: Yes, it is possible, but we didn't do it. We add this as a recommendation to the end of the conclusions.

Before:

This is necessary during the measurements, since raw data of single pulses are not stored and no postprocessing is possible.

Now:

This is necessary during the measurements, since raw data of single pulses **were** not stored and **thus** no postprocessing is possible.

End of conclusions:

**For conditions with low backscatter due to the low aerosol concentration as it is typical for the polar regions, a possibility to optimize the averaging time of the lidar would be the storage of the of the raw data (spectra) for post-processing.**

*COMMENT [RC2]: P6L26-27: If the PS96 data contain less noise then I would expect that the SNR is higher. How can this mean, that you need a different SNR threshold? This is a bit counterintuitive. Except the so-called "SNR" is the signal and not an actual SNR?*

The other Reviewer although raised this issue. We added the following clarification (page 6, line 25-27)

Before:

Additionally due to the different averaging time for each ray during PS85 and PS96 (1.5 vs 12–15 sec), the PS96 data contains less noise and thus it makes sense to choose a different SNR threshold for each data set.

After:

Additionally due to the longer averaging time for each ray during PS96 (12–15 sec) than during PS85 (1.5 sec), the PS96 data **allow for a lower SNR threshold compared to the PS85 data, because averaging over a longer period given the same SNR results in better data.** Thus, it makes sense to choose a **less strict** SNR threshold for the PS96 data **set to make both data sets more comparable**.

*COMMENT [RC2]: P6L27-30: Please explain why the vertical wind is around 0. This depends on the averaging period, on the precision of the angle of vertical stare (especially on a ship), and sometimes even on the synoptic situation (stationary waves, etc). In fact, you can determine the noise of your LOS wind by looking at the difference between the Autocovariance function at 0 and at the first leg. Or by evaluating the high-frequency tail of the wind power spectrum. And this could be done for different SNR thresholds.*

"The vertical wind is around 0 $ms^{-1}$ " just means it is close to zero. Päschke et al. (2015) found that this is valid for quiescent atmospheric conditions. Measured vertical velocities exceeding 5 $ms^{-1}$ can be safely considered as noise for the conditions of our measurements. Since we used the 75° elevation measurements (instead of vertical stares), we assumed LOS velocities of +- 10 $ms^{-1}$ unrealistic. We added the following clarification

Before:

Knowing that these had to be around 0 $ms^{-1}$ , the influence of noise could be evaluated. We follow a similar approach and evaluated the Doppler velocity from all individual rays for VAD scans with an elevation of 75°

Now:

Knowing that **vertical velocities are close to zero, Päschke et al. (2015) could**

evaluate the influence of noise from vertical stares for quiescent atmospheric conditions. As we did not have a stabilizing platform, the evaluation of the vertical stares is not possible because of the influence of horizontal wind on the signal. To circumnavigate this problem, we follow a similar approach and evaluated the Doppler velocity from all individual rays for VAD scans with an elevation of 75°

*COMMENT [RC2]: P7L1-12: this paragraph is a bit hard to understand. Can you explain why you do not use the goodness of the VAD fit to determine when a VAD delivered good and bad results?*

We computed the goodness of the VAD fit, but we did not include this in our analysis so far. Of course using the goodness of the VAD fit is also a valid error measure of the wind retrieval, which includes also the inhomogeneity of the wind field. We added the goodness of the fit in the theory section (page 6 line 5):

**As the system of equations is only solved approximately for a given a solution (u\*,v\*,w\*) we can define a measure for the goodness of the fit. Päschke et al. (2015) define the coefficient of determination. We define the fit deviation in our paper as:**

**<formula>** $||Matrix * (u^*, v^*, w^*) - Vector||_2$

**For the purpose of comparing the fit deviation, only scans with the same elevation should be used. It should also be noted that measuring a non-homogenous or non-stationary wind field would result in a larger fit deviation value.**

We also added the fit deviation to Figure 8 and 10 (Appended Fig. 1 and 2) and changed bottom/middle/top references to a/b/c/d in the text

New Figure Description:

**Lidar wind speed (a) and direction (b) for -25 dB SNR threshold for the 12 June 2014 (location see PS85 in Fig. 1). Colors below the black line (40 m) show the**

wind measurements of RV Polarstern (anemometer). The plot (c) presents the SNR thresholds that would allow for a wind calculation. The grey line is the cloud base from ceilometer measurements of RV Polarstern. The relative fit deviation (fit deviation divided by wind speed) is shown in (d). Values outside the colour range are plotted with the highest colour. Only scans with a 75° elevation where used.

We added the following text to page 9 line 4:

**The fit deviation (Fig. CITE, d) can help with this decision, but notice that the high relative fit deviation in the afternoon stems mostly from the low wind speeds.**

*COMMENT [RC2]: P7L5: It is mentioned that the VAD results are averaged for 20 min. But since it is possible that there are outliers in those VAD results shouldn't the median be a better indicator here?*

We agree that there are more options for the post-processing. This could include also a filtering of outliers in each vertical profile. In Figure 5 and Table 3 we want to distinguish the good/bad influence of choosing a different SNR threshold or other configurations and thus there is also an influence of outliers.

*COMMENT [RC2]: P7-8L20: The bias of 10° for the wind direction is quite unexpected. One would assume that radiosondes and the lidar both use GPS? Or is there a magnetic compass involved somewhere which might show a bias in Polar regions? It is a bit unsatisfactory that the reason for this bias remains unclear here. What is the VAISALA specification for their wind-direction bias?*

The bias is between 3 and 7° for the radiosonde comparison (not: 10°). Yes, the radiosondes use GPS. The lidar wind direction is computed relative to the ship and is modified by the data from the ship navigation system (as far as we know this is a combination of magnetic compass and GPS). We contacted the people maintaining

the ship navigation system to determine if there could be a bias (either with the system or along the way of data transmission and data base up-/download), but that seemed not to be the case. The reference system for the ship navigation system and the radio sounding is true north (not magnetic north). Vaisala states an uncertainty of 2° for their radiosondes (standard deviation of differences in twin soundings, wind speed above 3 m/s)

[We noticed the link for the Vaisala reference was not shown correctly and changed it]

**COMMENT [RC2]: P8L31: here it should be -17dB.**

Yes. We searched our paper for "dB" and also added missing minus signs in some other cases.

**COMMENT [RC2]: P9L5-15: The occurrence of these 3 fast LLJs is interesting. Can you give any explanation of the processes? Usually, in these latitudes, the typical Ekman oscillation time should be 12 hrs. So there must be another cause for these LLJs. Maybe you could reference some work on Arctic LLJ (e.g. Jacobson, ACP, 2013).**

There are different mechanisms for the formation of LLJs: 1) decoupling of layers in the SBL resulting in supergeostrophic winds by an inertial oscillation. 2) baroclinity causing vertical wind shear, 3) katabatic winds, 4) topographic channeling and local density flows, 5) LLJ due to a change in surface friction e.g. from rough sea ice to smooth water surfaces. We did not want to go into too much detail of dynamical processes for our case study. The LLJs on 17 January 2016 were measured during the passage of a synoptic front. In addition, the ship was located in a polynya the lee of a huge iceberg (A23A, size about 60kmx80km), which causes low-level baroclinicity. So baroclinicity seems to be the main reason.

We add the following text to P9L15:

**The dynamics of the LLJs were not studied in detail. They occurred during the passage of a synoptic front, when the ship operated in a polynya the lee of a huge iceberg (A23A, size about 60kmx80km). Baroclinicity is therefore a likely reason for the LLJs. While LLJs caused by inertial oscillations are frequent in the Weddell Sea during winter (Andreas et al. 2000), the observed jets during PS96 are comparable to the situation of the summertime Arctic Ocean, where Jakobson et al. (2013) find mostly baroclinic jets associated with transient cyclones.**

Andreas, E. L., Claffy, K. J., and Makshtas, A. P. 2000. Low-level atmospheric jets and inversions over the western Weddell Sea, Boundary-layer meteorology, 97, 459–486. https://doi.org/10.1023/A:1002793831076

Jakobson, L., Vihma, T., Jakobson, E., Palo, T., Männik, A., and Jaagus, J.: Low-level jet characteristics over the Arctic Ocean in spring and summer, Atmos. Chem. Phys., 13, 11089-11099 [Titel anhand dieser ISBN in Citavi-Projekt übernehmen] , https://doi.org/10.5194/acp-13-11089-2013, 2013.

*COMMENT [RC2]: P9L17: A bit general comment: I believe the topic of the paper should not be: The radiosondes are correct and let's see how well the lidar can be verified by this. A Doppler lidar by its design is one of the best methods to measure wind. In the end, it just depends on how precise one can measure frequencies. It is more a question of how much errors are generated by a moving platform like a ship. And what are the best scan strategies on a ship? It would be nice to get some answer to these questions, as other researchers might profit from this.*

Before:

We presented a verification of wind speed profiles measured by a wind lidar during two cruises of the research vessel Polarstern in the Arctic and Antarctic. The lidar was not motion-stabilized, but ship motions and the ship's orientation were measured by the

ship's navigation system and by a high-frequency Attitude Heading Reference System. The wind calculation is based on VAD scans with eight directions (rays), thus there is a high oversampling which allows for additional quality tests. [. . .] The comparison with the routine wind measurements of the ship at 40 m height yields a larger data set and a similar bias and RMSD.

Now:

We present a verification of wind speed profiles measured by a wind lidar **without a stabilizing platform** during two cruises of the research vessel Polarstern in the Arctic and Antarctic. The ship motions and the ship's orientation were measured by the ship's navigation system and by a high-frequency Attitude Heading Reference System. The wind calculation is generally based on VAD scans with eight directions (rays) **at an elevation angle of $75°$ (an elevation of $85°$ was discarded after a short test period)**, thus there is a high oversampling which allows for additional quality tests. **Wind retrievals from scans at multiple elevation angles elevation angles (25, 50 and $75°$) slightly improve the quality of the wind profile, but take more time. The low aerosol concentrations in polar regions result in a low backscatter. As a strategy to optimize the backscatter signal for these conditions the adjustment of telescope focal length of the lidar and the averaging time is useful.** [. . .] The comparison with the routine wind measurements of the ship at 40 m height yields a larger data set and a similar bias and RMSD. **The choice of a longer averaging time is preferred as it allows to reduce the SNR threshold and thus increases the amount of data. For longer averaging times the influence of the ship's movement can be higher, but this effect is small in our case because the ship operated mainly in sea ice where wave heights are relatively small.**

*COMMENT [RC2]: P9L27: "The RMSD is $10°$ but we also find $5°$". This sentence needs to be revised.*

Before:

For wind direction, the RMSD is about 10°, but we also find a bias of 5°. In conclusion, the results of the postprocessing of non-motion stabilized lidar data achieve comparable good quality as the motion-stabilized lidar study of Achtert et al. (2015).

Now:

For wind direction, the RMSD is about 10°, **which is comparable to other studies. The mean bias between radiosondes and lidar is about 5°. This is higher than the value of 2° found by Achtert et al. (2015), who find a bias of 5°only at higher levels, which is explained by the drift of the radiosonde and the resulting decrease in collocation of the measurements. Overall** the results of the postprocessing of non-motion stabilized lidar data achieve comparable good quality as the motion-stabilized lidar study of Achtert et al. (2015).

*COMMENT [RC2]: P10L1: "turning the wind perpendicular to the wind". Something is wrong here.*

Before:

Turning the wind perpendicular to the wind is desirable.

Now:

Turning the **ship** perpendicular to the wind is desirable.

*COMMENT [RC2]: Fig.3: Can this also be done for the YAW angle? It seems the wind direction error bias is highest. So probably one should have a look at this angle, too.*

Fig. 3 is focused on the roll and pitch angle during "in sea ice" condition. And shows that high frequency data is not that important but a low frequency data set already does good work. The yaw angle is another issue not focused on polar regions or motion-stabilizing. The 2 min median subtraction gives no relevant information in the yaw case.

So we plotted 15 second median subtraction (appropriate for our average time). See appended Fig. 3. We think the benefit of such a plot is too small as it just gives an approximation of how much data is removed by the sd(yaw) < 2° criterion. We instead added this information on page 4 line 25ff

Before:

It should be noted that the correction and filtering process causes almost no loss of data, as the ship's movement even during ice breaking conditions generally does not result in high-frequency changes of roll and pitch (except some cases of ramming).

Now:

It should be noted that the correction and filtering process causes almost no loss of data. **Only in 6% of the time, the standard deviation of the yaw angle over 15 seconds is larger than 2° and** the ship's movement even during ice breaking conditions generally does not result in high-frequency changes of roll and pitch (except some cases of ramming).

***COMMENT [RC2]: Fig.5: If Intensity is SNR+1, why does the axis start at 0.99?***

The way the lidar software computes the SNR (subtracting a background noise from the signal) can cause negative values for the signal (and thus a negative SNR). This question should be resolved by our changes concerning comment P5L6

***COMMENT [RC2]: Fig.7: It would be nice to include a correlation plot.***

We do not think the plot gives additional information (Appended Fig. 4 and 5). It is close to 1 except where there is not enough data with the same characteristics as the RMSD and bias. The numbers can better be seen in Table 2 and 3. We added the following sentence on page 7 line 24

Before:

Furthermore, a systematic dependence on height is not present. At heights above 1000 m the sample size is relatively small and differences between different SNR thresholds are not robust.

Now:

Furthermore, a systematic dependence on height is not present. **We also check for a height dependence of the correlation (not shown), but there was none present.** At heights above 1000 m the sample size is relatively small and differences between different SNR thresholds are not robust.

*COMMENT [RC2]: Fig.8: What is seen on the y-axis? Height? If it is height, how can there be measurements several hundred meters above the cloud base? Please explain in the text.*

Yes it is height, we added the label. We checked the backscatter data of the lidar found that it was not as high as for example after 1200 UTC. We appended the following explanation to the section 4.1 PS85 - Arctic 2014/06/12:

**Note that the height difference between lidar and ceilometer from 0800 to 1200 UTC is likely due to a thin layer of low clouds that the lidar could partially penetrate.**

*COMMENT [RC2]: Fig.9: Theta can be omitted and the x-axis can be annotated "POT.TEMPERATURE (°C)". Also, the date can be omitted since it is mentioned in the figure caption.*

All axis could be relabelled, so we did that (Appended Fig. 6 and 7). And added to the Figure description: "25, 50 and 75° elevation scans where used."

*COMMENT [RC2]: Table 1: The SNR threshold is -20dB, but in the text, it is written -18.2dB is from the manufacturer. So what means "default"?*

We just to give an approximate number in the table, but we agree that it is more confusing than helpful. We removed the row in Table 1

"Threshold for signal-to-noise ratio (SNR) : variable (default -20 dB)"

[Figure]

[Figure]

**Fig. 1.** added fit deviation to Fig. 8

a) wind speed (m s⁻¹)

b) wind direction (°)

c) SNR threshold (dB)

d) rel. fit deviation (%)

**Fig. 2.** added fit deviation to Fig. 10

[Figure]

**Fig. 3.** added yaw angle to Fig. 3

[Figure]

**Fig. 4.** added correlation to Fig. 7 for PS85

[Figure]

PS96

**Fig. 5.** added correlation to Fig. 7 for PS96

[Figure]

**Fig. 6.** changed labels of Fig. 9

[Figure]

**Fig. 7.** changed labels of Fig. 11

---

## Author Response (AR1)

**Authors response to Anonymous Referee #1 (Received and published: 18 July 2018)**

The paper by Zentek et al. describes the use of a scanning lidar for ship-borne wind measurements without a motion stabilisation platform. The authors used data collected with an external Altitude Heading Reference System to correct for the ship's pitch and roll after the measurement campaign. The presented technique and the statistical comparison of the lidar wind measurements to radio soundings as well as to ship measurements is important for the scientific community due to the clear need for wind measurements over the oceans – especially in the polar regions. Such measurements are important for a better understanding of atmospheric processes in the maritime environment. The paper is suitable for publication in AMT and can be published after minor revision.

**Major comment:**

- The lidar measurements that have been corrected for the ship's pitch and roll after the measurements are performed consist of profiles that are the average of 12 to 15 seconds of individual rays for the PS96 campaign and 1.5 seconds for the PS85 campaign. The movement of the ship during these averaging periods introduces horizontal wind components into the vertical wind. This is an important source of error and should be discussed in the paper. How does the proposed methodology account for movements during the time needed to obtain the averaged profiles that are later motion corrected?

We address this issue in section 2.3.1 "Ship motion correction" (page 4, line 23ff & Fig.3) but never stated clearly what we address and just refer to it as "the error" (page 4, line 25). We corrected this

Before:

During PS96 the averaging time of a single ray was typically 12–15 seconds, so that we corrected each single measurement with the mean value over the averaging time. All measurements that have a standard deviation of roll or pitch angle larger than 0.5° or yaw angle larger than 2° over this averaging time were excluded from the analysis in order to reduce the error.

[... other text ...]

For a data point in 1 km distance from the lidar a change of elevation from 75° to 75.5° (25° to 25.5°) causes a difference in height of 2 m (8 m) and a horizontal wind speed error of less than 3.3% (0.4%). This is acceptable as we will later interpolate over height intervals of 50 m.

Now:

During PS96 the averaging time of a single ray was typically 12–15 seconds, so that we corrected each single measurement with the mean value over the averaging time. This introduces an error whenever the ship's angle and thus the lidar angle changes during this averaging time. In order to reduce the error, all measurements that have a standard deviation of roll or pitch angle larger than 0.5° or yaw angle larger than 2° over this averaging time were excluded from the analysis. Correcting the direction of the lidar measurement by the mean roll and pitch angle during the averaging time should already cause most of the error to average out, as it measures partly too much and partly too less wind speed. But even if this is not the case, for a data point in 1 km distance from the lidar a change of elevation from 75° to 75.5° (25° to 25.5°) causes a difference in height of 2 m (8 m) and the resulting horizontal wind speed error is less than 3.3% (0.4%). This is acceptable as we will later interpolate over height intervals of 50 m and only evaluate the horizontal wind in our paper.

Other comments:

- Page 5, line 12: Can you really assume horizontal homogeneous wind fields? The elevation changes during the scan.

We assume the homogeneity for a fixed height and only take data points from that height (page 5, line 10f). A different elevation as such is no problem. For example the wind velocity could still be computed if there are 3 different measurements with elevations of 50, 60 and 70° and azimuths of 20, 30, and 40°

The error resulting from the change of elevation (due to the ships movement) during each single ray was discussed in section 2.3.1.

- Page 6 line 30: Doppler velocity due to horizontal wind speed is less than 26 % at this elevation. Is that still true if you correct pitch and roll after the measurements were taken? Your elevation is not stable at 75° due to the ship's motion.

Yes, a Doppler velocity of 10 m s^-1 for an elevation of 73/75/77° would result in 34/39/44 m s^-1 horizontal wind speed. These values can be safely considered to be unrealistic for our conditions.

- Page 7 line 9-10: What are the reasons for the different SNR thresholds for the two campaigns? Could it be the different averaging times of the rays? The elevation is not stable during the measurements and you get different horizontal wind components into your vertical wind component. With a longer averaging time the effect might be enhanced.

The reviewer is partially correct. As stated on page 6 line 15, the value for a SNR threshold can vary depending on the instrument specific performance (detector noise) and the variability of atmospheric conditions within the measured volume. We think that the main reason is not the influence of elevation but the averaging time itself. We changed the passage explaining this. Before:

"Additionally due to the different averaging time for each ray during PS85 and PS96 (1.5 vs 12–15 sec), the PS96 data contains less noise and thus it makes sense to choose a different SNR threshold for each data set." (page 6, line 25-27)

After:

"Additionally due to the longer averaging time for each ray during PS96 (12–15 sec) than during PS85 (1.5 sec), the PS96 data allow for a lower SNR threshold compared to the PS85 data, because averaging over a longer period given the same SNR results in better data. Thus, it makes sense to choose a less strict SNR threshold for the PS96 data set to make both data sets more comparable."

- Table 2 and Table 3: Similar to previous comment the statistics for the PS85 campaign with a shorter averaging time are better than for PS96 with a longer averaging time. What is the reason for this?

We think our data sets are too small to reach any definite conclusion or even to be reasonably certain that one data set / scanning technique is really better than the other. We think it is possible, that external sources like a bias in weather condition during one cruise could be reason enough to cause this differences. One reason to include Table 3 in this paper was to show that a different reasonable analysis-configuration would lead to different statistics. For example the computed RMSD for PS96 can change from 0.9 to 0.7 m s^-1 making it the same as for PS85.

- Page 8 line 19/20: Could the higher bias be explained by not having a horizontally homogeneous wind field? You only correct for the elevation and azimuth but you cannot correct for the horizontal wind component being present in the vertical wind component. We see no reason why this would lead to a positive bias in wind direction for both cruises and for both comparisons (radio sounding and anemometer).

- Figure 6: please add a plot for the relative difference between lidar and radio soundings by height for wind speed and wind direction.

We do not know of any definition for relative differences of wind directions. (Just labeling the axis differently by dividing by 180°?)

We plotted the relative difference for wind speed and absolute difference for wind direction with the symbol "+" for each single case. We also plotted the mean relative difference for wind speed as lines scaled with a factor of 4.

We don't think that the benefit is high enough to add this additional plots to the paper (But we change the Figure 6 by adding a little space between the RMSD and bias subplots.)